# Severe Early-Onset Intrahepatic Cholestasis of Pregnancy Following Ovarian Hyperstimulation Syndrome with Pulmonary Presentation after In Vitro Fertilization: Case Report and Systematic Review of Case Reports

**DOI:** 10.3390/life14010129

**Published:** 2024-01-16

**Authors:** Stipe Dumančić, Mislav Mikuš, Zdenka Palčić, Dubravko Habek, Mara Tešanović, Marko Dražen Mimica, Jelena Marušić

**Affiliations:** 1Department of Obstetrics and Gynecology, Clinical Hospital Center Split, 21 000 Split, Croatia; stipe.dumancic@gmail.com (S.D.); mdmimica@kbsplit.hr (M.D.M.); 2Department of Obstetrics and Gynecology, Clinical Hospital Center Zagreb, 10 000 Zagreb, Croatia; 3School of Medicine, Catholic University of Croatia, Ilica 242, 10 000 Zagreb, Croatia; dhabek@unicath.hr; 4Department of Obstetrics and Gynecology, General Hospital Dubrovnik, 20 000 Dubrovnik, Croatia; mara.tesanovic56@gmail.com; 5School of Medicine, University of Split, Soltanska 2, 21 000 Split, Croatia; 6University Department of Health Studies, University of Split, R. Boskovica 35, 21 000 Split, Croatia

**Keywords:** early-onset intrahepatic cholestasis of pregnancy, in vitro fertilization, ovarian hyperstimulation syndrome, pleural effusion, ursodeoxycholic acid

## Abstract

Background: Intrahepatic cholestasis of pregnancy (ICP) is the most common pregnancy-related liver disease, usually presented in the third trimester with pruritus, elevated transaminase, and serum total bile acids. Evidence shows that it can be developed in the first trimester, more commonly after in vitro fertilization (IVF) procedures, with the presence of ovarian hyperstimulation syndrome (OHSS). Methods: A literature search was conducted in the PubMed/MEDLINE database of case reports/studies reporting early-onset ICP in spontaneous and IVF pregnancies published until July 2023. Results: Thirty articles on early-onset ICP were included in the review analysis, with 19 patients who developed ICP in spontaneous pregnancy and 15 patients who developed ICP in IVF pregnancies with or without OHSS. Cases of 1st and 2nd trimester ICP in terms of “early-onset” ICP were pooled to gather additional findings. Conclusions: Proper monitoring should be applied even before expected pregnancy and during IVF procedures in patients with known risk factors for OHSS and ICP development (patient and family history), with proper progesterone supplementation dosage and genetic testing in case of ICP recurrence.

## 1. Introduction

Intrahepatic cholestasis of pregnancy (ICP) is a rare condition usually presented in the 3rd trimester with pruritus gravidarum and elevated liver function tests (LFTs), transaminases in particular, and serum total bile acids (TBAs). However, emerging evidence shows ICP to be a major obstetric issue developing as early as the first trimester, as it is not a preferred differential diagnosis of pruritus gravidarum in early pregnancy [1]. Clinical presentation of early-onset ICP, compared to late-onset ICP, includes earlier onset of the disease, long disease duration with recurrence during pregnancy, and a higher incidence of preterm labor, fetal distress, and lower birth weights compared to late-onset ICP [2,3,4]. Mutlu et al. proposed that the early-onset ICP in the first trimester might be more appropriate to name “ICP-like disease” than ICP since it is typically seen in the last trimester of pregnancy with a slightly different etiology related to assisted reproductive technology (ART) and consequent OHSS [5,6]. The study of Lin et al. proposed 34 weeks of gestational age as the cut-off level for differentiating early and late-onset ICP based on serum total bile acid and glycocholic acid pre-treatment concentrations, which have been shown to be significantly higher with early-onset ICP [2]. The cause of intrahepatic cholestasis is still unknown, but it is suspected to be related to higher sex steroid hormone levels, with hyperestrogenism as the basis of early-onset ICP development. Other risk factors for early-onset ICP development could be contraceptive usage, ovarian hyperstimulation with OHSS development, multiple pregnancies, and genetic predisposition with mutations of genes involved in hepatocellular bile acid transport mechanisms.

ICP is rarely observed during the extremely early pregnancy stage, with only a few cases reported in the literature. Herein, we report severe early-onset ICP that developed in 32-year-old primigravida at 5 + 3/7 weeks of gestational age, first observed as acute hepatic injury, but with suspicion of ICP due to the TBAs elevation. The patient first presented with the OHSS, with a pulmonary presentation, i.e., pleural effusion and progressive dyspnea. She was conceived using assisted reproductive technology (ART) techniques by IVF/intracytoplasmic sperm injection (ICSI) due to idiopathic infertility. Starting from the ART procedure to the delivery of dichorionic diamniotic twins in the 36th gestational week, she was followed up and treated by specialists in the University Hospital Center (UHC) Split. The aim of this case report is to showcase patient management throughout IVF treatment and pregnancy, as our early-onset ICP case is one of the most difficult cases reported in the literature. Previous studies published in the literature usually analyzed patients with ICP in the late 2nd or 3rd trimester. Regarding the scarcity of prospective data observing early-onset ICP with/without OHSS, we conducted a systematic literature review of case reports presenting early-onset ICP in the 1st and 2nd trimesters of spontaneous or IVF pregnancies to provide additional data on early-onset ICP.

## 2. Materials and Methods

### 2.1. Patient Consent and Case Report

The patient signed informed consent to allow data collection (from the electronic health records of UHC Split) for research purposes and to allow publication of the case report. Furthermore, we obtained ethical approval from the Ethics Committee of the UHC Split to use patient data for research purposes. This case report is written according to the Case Reports (CARE) guidelines and checklist [7].

### 2.2. Systematic Review of Case Reports

#### 2.2.1. Search Strategy and Eligibility Criteria

A systematic literature search of case reports was performed according to the Preferred Reporting Items for Systematic Reviews and Meta-Analysis (PRISMA) guidelines [8]. We searched for relevant English-written case reports or case studies in the PubMed (Medline) database until July 2023 using the selected terms: “intrahepatic cholestasis of pregnancy”, “liver failure”, “spontaneous pregnancy”, “in vitro fertilization”, “ovarian hyperstimulation syndrome”, “early onset” or “1st trimester”, and/or corresponding Medical Subject Headings (MESH) terms, using AND/OR operators. Authors (SD, MM, MT) screened abstracts and reviewed selected case reports for full-text analysis, with disagreements resolved by discussion with other authors (MJ, MDM, DH). Case reports and/or studies that reported early-onset ICP or liver failure in spontaneous or IVF pregnancy were included for full-text article analysis. Furthermore, we also reviewed gray literature (conference abstracts) and references to the included articles to identify other relevant articles. Exclusion criteria were other types of non-English primary studies and secondary publications, case reports or studies that reported individual cases of intrahepatic cholestasis in pregnancy or OHSS, reports of mild liver function tests’ aberrations in pregnancy, reports of early-onset ICP with insufficient data, cholestasis in non-pregnant women, or in males. 

#### 2.2.2. Data Extraction and Outcome Measures

From the included manuscripts, we extracted patient age, ethnicity, gravidity and parity, IVF indication, number of embryo transfers (ET), OHSS severity, history of liver dysfunction prior to pregnancy related to ICP, presence of jaundice, ICP onset and relapse during pregnancy, type of ICP treatment, peak liver function tests’ (LFT) values, pregnancy outcomes with gestational age (GA) at delivery, gestation, live- and stillbirth, delivery modes, and obstetrical and perinatal complications. In the case of different measuring units for LFTs, we used online calculators to standardize data collection. In cases of missing OHSS grades, we classified OHSS severity as mild, moderate, or severe based on the data reported in case reports. 

The main outcomes of this systematic literature review are:(1)clinical presentation of early-onset intrahepatic cholestasis of pregnancy (ICP) following ovarian hyperstimulation syndrome (OHSS),(2)clinical differences between early-onset intrahepatic cholestasis in spontaneous versus IVF pregnancies,(3)early-onset ICP treatments,(4)delivery and perinatal outcomes with early-onset ICP.

#### 2.2.3. Statistical Analysis

Categorical data are presented as numbers and percentages, while continuous data are presented with a mean (standard deviation) or median (minimum-maximum). The differences between groups were tested by the χ^2^ test or Mann–Whitney *U* test, respectively. The *p*-value was set to 0.05. Statistical analysis was carried out using JASP Team (2023, Version 0.17.2.1) software.

## 3. Results

### 3.1. Case Presentation

This is a 32-year-old Caucasian primigravida without prior spontaneous abortions. She underwent hysteroscopic ablation of an endometrial polyp a year prior to this pregnancy. She could not conceive spontaneously for 4 years, with the diagnosis of idiopathic infertility. She was an otherwise healthy woman, without liver disease, with an anti-Müllerian hormone (AMH) level of 18.5 pmol/L. Her pre-pregnancy body mass index (BMI) was 20.81 kg/m^2^. However, she reported a family history of liver problems, in particulara the intrahepatic cholestasis of pregnancy her mother and sister experienced. Her sister was hospitalized in the 32nd gestational week due to elevated aminotransferase levels (above 1000 U/L), with the delivery by emergency cesarean section in the 33rd gestational week. Due to the family history, our patient underwent diagnostics for liver diseases without a proven disease.

The patient underwent a controlled ovarian hyperstimulation (COH) process: she was stimulated with 175 IU Gonal-F^®^ (Merck KgaA, Darmstadt, Germany) from the 2nd to the 12th cycle day, within a flexible gonadotropin-releasing hormone (GnRH) antagonist protocol with Cetrotide^®^ (Merck KGaA, Darmstadt, Germany) (0.25 mg) starting from the 8th cycle day. A follicular study showed 12 ovulatory dominant follicles, with 5 in the right ovary and 7 in the left ovary, with diameters between 15 and 19 mm and an endometrial thickness of 12 mm. She received a human chorionic gonadotropin (hCG) trigger shot on the 12th cycle day (Ovitrelle^®^ 250 μg, Merck Global, Darmstadt, Germany), when final maturation was assessed after 36 h, with 11 oocytes obtained by ultrasound-guided aspiration. She underwent an 8-cell double embryo transfer (day 2) with routine progesterone support with 600 mg/day Utrogestan^®^ (Cyndea Pharma S.L., Ólvega, Spain) vaginal capsules.

On the 8th day after ET, the patient presented to the emergency gynecological service due to cough, orthopnea, chest pressure, and distended abdomen, under the clinical presentation of ovarian hyperstimulation syndrome with the pulmonary component. Ultrasound showed enlarged ovaries bilaterally (circa 8 × 7 cm and 8 × 6 cm), with 400 mL effusion in the Douglas pouch. Before and during COH, the patient did show risk factors for OHSS (i.e., patient’s age, BMI, AMH, antral follicle count—AFC, antagonistic protocol, number of follicles during routine ultrasound monitoring, number of oocytes, Douglas pouch without free fluid), and upon admission without laboratory signs of hemoconcentration or lower albumin levels, with beta-hCG of 121.0 IU/L (Figure 1). Following hospitalization, a therapeutic right thoracocentesis was indicated, with 1000 mL of sanguineous pleural effusion relieved. Dyspnea persisted for the next two days, but a chest ultrasound showed a low pleural effusion level without the need for thoracocentesis. However, on the second day upon admission, an increase in aminotransferase values was first observed: aspartate aminotransferase (AST) 99U/L and alanine transaminase (ALT) 101 U/L, with elevated total bilirubin without jaundice. Figure 1 presents the trendline of laboratory values during serial serum laboratory tests.

On the 4th day of hospitalization, a further increase in liver enzymes was recorded, along with the cessation of respiratory difficulties. However, on the 8th day of hospitalization, aminotransferase values were AST 750 U/L and ALT 683 U/L, without high gamma-glutamyl transferase (GGT) elevation, with a normal albumin level and beta-hCG 1506.0 IU/L. Transvaginal ultrasound showed a 14 mm endometrium with 300 mL effusion in the Douglas pouch. Due to the disturbed hepatogram, despite improvement in the patient’s condition, an abdominal ultrasound was indicated. The study showed normal size, liver echogenicity, perihepatic, and Morison’s pouch-free fluid. On the ninth day upon admission, the patient developed jaundice (scleral) with gastrointestinal problems. Transaminases were still increasing (AST 1072 U/L, ALT 1083 U/L), with increased bile acids (44 μmol/L) and beta-hCG (2177.0 IU/L). Transabdominal ultrasound again showed normal morphology with persistent perihepatic free fluid. That day, the patient discontinued using Utrogestan^®^ vaginal capsules and prophylactic vitamin preparations. 

Due to the short period of decline and the re-increasing in conjugated bilirubin and bile acid values (bile acids up to 93 μmol/L without observed GGT and alkaline phosphatase ALP elevation), intrahepatic cholestasis was suspected on the 13th day of hospitalization, and the patient began treatment with ursodeoxycholic acid (UDCA, Ursofalk^®^, DR. FALK PHARMA GmbH, Freiburg, Germany) at 250 mg twice daily without the inclusion of proposed prednisolone. Indicated serology testing did not confirm viral nor autoimmune hepatitis (except for borderline antinuclear ANA antibody value) and normal ceruloplasmin value. With the UDCA therapy, the patient’s clinical condition was better, which was followed by the decline of elevated values of liver transaminases, bilirubin, and bile acids. On the 21st day of hospitalization, a transvaginal ultrasound showed a dichorionic diamniotic twin pregnancy with positive heartbeats in both embryos.

UDCA therapy was continued until delivery, and the patient was followed up by gastroenterologists with routine LFT check-ups and monitored regularly by perinatologists. She was hospitalized again at the 25th gestational week due to ICP and suspicion of threatened premature labor with vaginal bleeding due to placenta previa. Liver enzymes had borderline values. After the symptoms subsided, she was discharged and sent home. The pregnancy was completed by emergency Caesarean section at 36 + 3 weeks of gestation due to premature prelabor rupture of membranes (PPROM). The first gem delivered was male, with a gestational weight of 2730 g and and a height of 47 cm, with an Apgar index of 1st minute 10. The second gem was a female child with a gestational weight of 2310 g and 45 cm in height, with the Apgar index in the 1st minute 9. The placentas were sent for pathohistological analysis, which showed normal findings.

### 3.2. Systematic Review of Case Reports

#### 3.2.1. Literature Search and Data Extraction

The identification of articles reporting early-onset ICP in spontaneous or IVF pregnancies in the form of a PRISMA diagram is shown in Figure 2. After the abstract screening, we considered 52 articles reporting early-onset intrahepatic cholestasis of pregnancy (eICP) in spontaneous pregnancy or following ovarian hyperstimulation syndrome (OHSS) to be eligible for full-text article analysis. However, after full consideration, we excluded 22 articles based on the following criteria: articles reporting other liver diseases not related to ICP (*n* = 9), cases of 3rd trimester ICP (*n* = 9), or cases of ICP without sufficient data to be included in the review (*n* = 4). Thirty articles reporting 34 patients with ICP are included in this review analysis, with 19 patients developing ICP in spontaneous pregnancy, while 15 patients developed ICP in IVF pregnancies with or without OHSS [5,6,9,10,11,12,13,14,15,16,17,18,19,20,21,22,23,24,25,26,27,28,29,30,31,32,33,34,35,36]. Furthermore, we pooled cases of 1st and 2nd trimester ICP (11 cases) in terms of “early-onset” ICP to gather additional findings. An analysis summary of the systematic review is presented in Table 1.

#### 3.2.2. First Trimester ICP following OHSS

Sixty-six percent of IVF pregnancies were complicated by OHSS during the 1st trimester and were followed by ICP development. These were moderate to severe OHSS cases, with only 1 mild OHSS case. Two IVF patients who developed ICP in 2nd trimester were not complicated by OHSS. We can observe that these were primigravids, with primary infertility in most cases, and there was a median of double ET. Furthermore, 3 cases of twin pregnancies from this group were all complicated by OHSS, and 2 of 3 cases with ICP recurrence during pregnancy were complicated by OHSS. Compared to the LFTs reported in Table 1, OHSS pregnancies had slightly higher median values of transaminases and TBAs (ALT: 369, AST: 311, TBA: 166.8). Cases of miscarriages were complicated by OHSS.

#### 3.2.3. Clinical Differences between Spontaneous vs. IVF Pregnancies with 1st Trimester ICP

From this systematic review, we can observe differences in clinical presentation between spontaneous and IVF pregnancies complicated by ICP in the 1st trimester (Table 1). Patients were of similar age; however, observed differences in ethnicity were not statistically significant. Women in the spontaneous group were secundigravids or multiparas (*p* = 0.022, *p* = 0.006), compared to primigravids in the IVF group, with the majority of women diagnosed with primary infertility. Seventy percent of women in the spontaneous group reported a positive history related to ICP, compared to 15.4% in the IVF group (*p* = 0.008). Major medical issues developed in patients and their family members were ICP in previous pregnancy, contraception-induced cholestasis, and cholelithiasis with cholecystectomy. Furthermore, ICP had developed circa 3 weeks earlier in IVF pregnancies than in spontaneous pregnancies (*p* = 0.008), but with a similar jaundice occurrence or ICP recurrence in pregnancy. Due to insufficient data, we only reported several LFTs in this review. We observed higher transaminase values in the IVF group, while total bilirubin and bile acids were higher in the spontaneous group. However, the difference in LFT values was not statistically significant.

#### 3.2.4. Treatment of 1st Trimester ICP

From this review, we observed several medications used in the ICP treatment as followed by frequency: ursodeoxycholic acid (UDCA), antihistamines (diphenhydramine, hydroxyzine, clemastine, chlorpheniramine, and promethazine), corticosteroids (hydrocortisone, dexamethasone, betamethasone, and prednisolone), cholestyramine (as bile acid sequestrant), hypnotics (phenobarbital, zolpidem tartrate), S-adenosyl methionine (SAMe), and plasmapheresis as other therapeutic interventions. No statistically significant differences were observed between groups (Table 1).

#### 3.2.5. Pregnancy Outcomes with 1st Trimester ICP

In pregnancies complicated by 1st trimester ICP, there were 18 delivered newborns (9 in each group), with 4 miscarriages in the IVF group and one artificial abortion in spontaneous pregnancy. Deliveries were earlier in spontaneous pregnancies than in IVF ones, with most spontaneous pregnancies being preterm delivered. Only three cases of twin pregnancies were from IVF pregnancies. However, there was one case of stillbirth in the spontaneous pregnancy group. Regarding delivery modes, in the majority of cases, there were more vaginal than cesarean deliveries. Most spontaneous pregnancies were delivered by induced vaginal birth, while in the IVF group, there were more spontaneous vaginal deliveries. No statistically significant differences were observed between groups (Table 1). Due to preterm deliveries, PPROM was the most frequent reason for induced delivery, complicated by fetal distress or meconium-stained amniotic fluid (MAF) (*p* = 0.022). Other complications were as follows: LFT aberrations, a few cases with fetal distress, breech or transverse presentation, chorioamnionitis, oligohydramnios, and Doppler aberrations. Perinatal complications were observed only in the spontaneous group (*p* = 0.021), with respiratory distress being most prominent (*p* = 0.034).

#### 3.2.6. Pooled 1st and 2nd Trimester ICP Cases Regarding Early-Onset ICP

Analysis of pooled 1st and 2nd trimester ICP data showed similar results to 1st trimester ICP data (Table 1). Differences were shown to be significant regarding age (*p* = 0.048), ethnicity (*p* = 0.015), and a higher rate of preterm delivery in the spontaneous group (*p* = 0.035).

## 4. Discussion

Early-onset ICP after an IVF procedure and OHSS presented here is one of the most extreme cases compared to those published to date, with TBAs up to 166 μmol/L, AST 1202 U/L, and ALT 1545 U/L. Moreover, it is a unique case as the OHSS was presented with unilateral pleural effusion. A recent systematic review by Irani et al. concluded that OHSS can present with symptomatic pleural effusion without moderate to large ascites [37]. We closely monitored effusion status and intervened, when necessary, with thoracocentesis, which relieved the patient’s dyspnea. Our treatment showed efficient results since the OHSS and ICP were diagnosed early and treated successfully, and pregnancy continued without major fetal or maternal adverse events.

Before and during the COH treatment, our patient did not show risk signs for OHSS development, so we did not routinely monitor estrogen concentrations, which in turn could show the insidious development of OHSS. The primary prevention of OHSS is the identification of patients with risk factors and the adjustment of stimulation protocol accordingly, which we have conducted. As a standard clinic practice, routine estradiol monitoring is conducted only in patients with risk factors for OHSS. The ovulation trigger shot with human chorionic gonadotropin (hCG) is considered the main driver of OHSS development. Using a GnRH agonist instead of hCG as a trigger shot is another preventable measure. We also acknowledge that the progesterone supplementation in our patient, as prescribed following IVF treatment for progesterone support, could lead to the development of ICP in early pregnancy. The only risk factor for ICP in our patient was a family history of 3rd trimester ICP in her sister. Therefore, in cases of reported risk factors for ICP and OHSS, we advise using the lowest progesterone dose, which should be discontinued in cases of suspected early-onset ICP, even before a confirmed viable pregnancy.

Our systematic review yielded additional data for the investigation of early-onset ICP. Data regarding 1st trimester ICP are relatively similar to previous evidence [3,38]. Moreover, patient history and OHSS have been shown to be major risk factors. However, some contrary findings regarding LFT values and adverse events depict early-onset ICP as more severe in spontaneous pregnancies. However, these results show limits due to the retrospective nature and non-disclosed data within the included manuscripts, without supportive meta-analysis.

The cause of intrahepatic cholestasis is still unknown, but it is suspected to be related to sex steroid hormone levels as well as genes encoding hepatocellular transport mechanisms. Most of the early-onset ICP cases so far are from IVF pregnancies with ovarian hyperstimulation syndrome, after which pruritus develops with elevated liver enzymes and bile acids. The role of COH and subsequent OHSS in the pathogenesis of early-onset ICP could be that the corpus luteum formed during COH produces higher concentrations of estrogens, progesterone, and vasoactive mediators, with hyperestrogenism as the basis of early-onset ICP pathophysiology [39,40]. Similarly, higher estrogen production develops during multiple pregnancies with or without concomitant ICP [41]. Nonetheless, sulfated progesterone metabolites, raised in ICP patients, inhibit the farnesoid X receptor (FXR) mechanism for bile acid clearance and inhibit hepatocytes protein natrium-dependent thaurocholate cotransporting polypeptide (NTCP, variant SLC10A1) and canalicular bile salt export pump (BSEP), causing hepatocyte retention and elevation of bile acids, with the basis of adverse fetal events pathogenesis in ICP [42,43,44]. The study by Bolukbas and colleagues compared spontaneous and IVF pregnancies with ICP. They found higher TBAs in IVF pregnancies, with a higher number of multiple gestations, but fertilization type did not affect the onset of ICP symptoms [3]. Moreover, Feng et al. found that IVF pregnancies yielded a higher incidence of early-onset ICP (before 28 gestational weeks), a more frequent ICP symptomatology, and a higher rate of adverse obstetric events compared to spontaneous pregnancies [45]. However, Alemdaroglu et al. did not find any differences between IVF and control cases of singleton gestations in terms of clinical presentation and adverse fetal events related to ICP [38]. Hu et al. prospectively investigated the effect of moderate-to-critical OHSS on obstetric and perinatal outcomes. They found that pregnancies from IVF/ICSI and fresh ET procedures, complicated with OHSS, yielded a higher incidence of gestational diabetes mellitus (GDM), venous thromboses, and neonatal intensive care unit (NICU) admissions without effect on ICP development [46]. Researchers argue whether COH is safe in hepatic patients and if it can disturb hepatic function in IVF pregnancies, with the risk of ICP developing. A prospective cohort of 422 women by Romito et al. demonstrated that COH, by long GnRH-antagonist protocol, did not provoke renal and/or hepatic function aberrations, suggesting a good safety profile for COH [47]. Moreover, Rahim et al. showed that patients with chronic liver diseases and liver transplantation in a compensated state who underwent IVF procedures did not have higher rates of OHSS and ICP. However, increased ICP risks should be mentioned [48]. Finally, our systematic review revealed that early-onset ICP developed irrespective of fertilization type or OHSS presence, despite the highest rates in IVF pregnancies with OHSS. Some women present with a genetic predisposition for cholestasis in a hyperestrogenic state, like cases of contraceptive-induced cholestasis. Genetic studies showed that mutations of multidrug resistance protein 3/ATP-binding cassette subfamily B member 4 (MDR3, ABCB4) variations and canalicular bile salt export pump (BSEP, ABCB11) gene, which are involved in the hepatocellular mechanism of bile excretion, could have a role in the early-onset ICP pathogenesis [49,50].

Intrahepatic cholestasis is associated with adverse pregnancy outcomes, which are most often associated with high serum bile acid values (>40 mmol/L), including MAF, premature birth, fetal distress, and rarely sudden fetal death. Animal studies have shown that high values of bile acids induce myometrial contractions and premature birth, stimulate fetal intestinal motility with MAF, show effects on alveolar surfactant leading to lung failure and perinatal mortality, and affect cardiomyocytes leading to fatal arrhythmias, resulting in increased perinatal mortality [1,51,52]. A French study by Labbe et al. showed that early-onset ICP resulted in higher rates of adverse fetal events, predominantly threatening preterm birth and prematurity [53]. Moreover, Estiu et al. found that the risk of MAF is higher at the level of hyercholanemia at ICP diagnosis and earlier stages of ICP [4]. Glantz et al. showed that TBAs serve as an independent predictor of adverse fetal events if the concentration exceeds 40 μmol/L [54]. Besides TBAs, Jin et al. also found ALP and GGT as predictors of adverse fetal events in the early-onset ICP rather than other LFTs, given that higher ALP is associated with acute placental damage and preterm delivery [55]. Finally, Zhang et al. proposed a classification and regression tree (CART) model for the prediction of early-onset ICP in the first 20 gestational weeks based on the following parameters: ALT, GGT, fibrinogen (Fg), platelet large cell ratio (P-LCR), activated partial thromboplastin time (APTT), lactate dehydrogenase (LDH), creatinine (Cr), and mean corpuscular hemoglobin concentration (MCHC), with a sensitivity of 72.41%, a specificity of 79.69%, and a positive predictive value of 76.36% [56]. 

Pregnancies complicated by intrahepatic cholestasis are usually terminated by 36 weeks of pregnancy to reduce possible perinatal complications. Lin et al. found that inducing labor after 37 weeks can significantly improve perinatal outcomes, given the effective treatment is initiated after the diagnosis of early-onset ICP [2]. The study of 155 Latina women by Lee et al. showed that preterm labor does not prevent the occurrence of adverse fetal events [57]. 

## 5. Conclusions

In conclusion, our patient could serve as aa literature example of the management of pregnancy complicated by a plethora of disorders (OHSS with pleural effusion, early-onset ICP, PPROM), with only a positive familial history as a risk factor for ICP. Regarding the management of early-onset ICP, as we showed in our case report, and in cases of evident predisposed signs for ICP (COH/IVF procedures, positive history of risk factors, etc.), strict monitoring should be applied even before expected pregnancy. Before and during COH, it should be mandatory to monitor estradiol levels, which we omitted due to missing risk factors for OHSS. Proper conservative therapy should be applied, with discontinuation of progesterone support after IVF procedures in cases of early-onset ICP suspicion and UDCA initiation as soon as diagnosed, with daily hepatogram monitoring. Marin et al. concluded that screening for genetic polymorphisms of ABCB4 in pregnancy is neither cost-effective nor evidence-based [6]. Genetic testing in isolated cases of early-onset ICP could be beneficial as a confirmatory method and supplementary guideline since the ICP is considered a recurrent disorder in future pregnancies. It is necessary to further investigate the delivery of pregnancies complicated by early-onset ICP, labor induction, or Caesarean delivery.

## Figures and Tables

**Figure 1 life-14-00129-f001:**
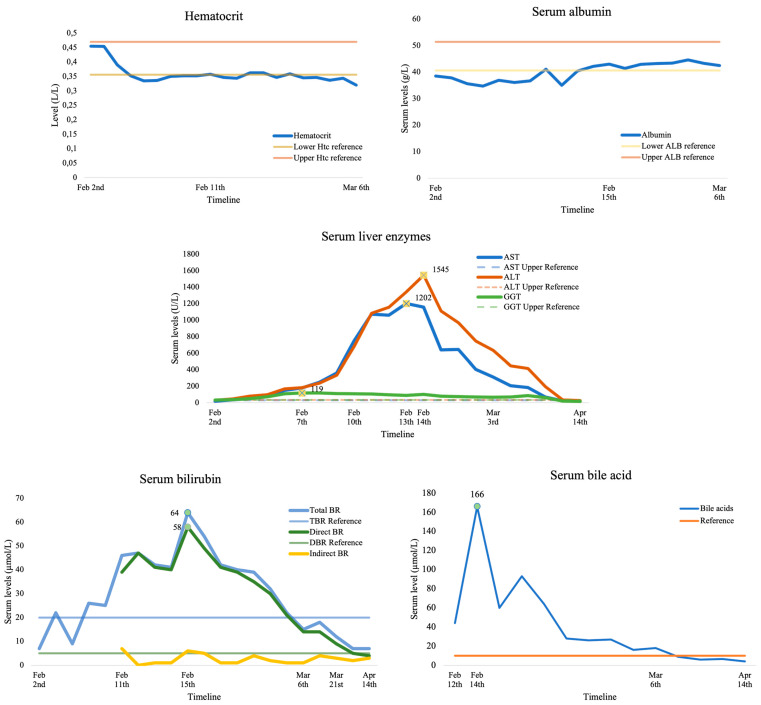
The trendline of laboratory values during serial serum laboratory tests.

**Figure 2 life-14-00129-f002:**
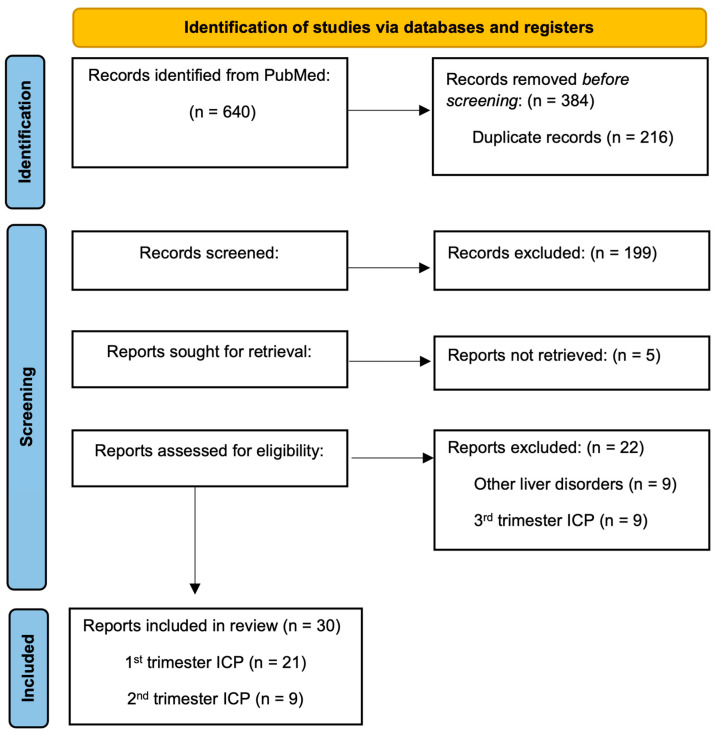
PRISMA diagram.

**Table 1 life-14-00129-t001:** Review summary of early-onset ICP cases in spontaneous versus IVF pregnancies.

	Early-Onset ICP		*p*	1st Trimester ICP		*p*
	Spontaneous (*n* = 19)	IVF (*n* = 15)		Spontaneous (*n* = 10)	IVF (*n* = 13)	
Age, years	28.2 ± 4.4	30.9 ± 3.2	0.048	29 ± 4.8	31.1 ± 3.4	0.233
Ethnicity			0.015			0.059
Caucasian	15 (78.9)	5 (33.3)		8 (80)	4 (30.8)	
Asian	1 (5.3)	5 (33.3)		0	4 (30.8)	
Hispanic	2 (10.5)	0		1 (10)	0	
Middle Eastern	1 (5.3)	3 (20)		1 (10)	3 (23.1)	
Indian	0	2 (13.3)		0	2 (15.3)	
Gravidity	2 (1–6)	1 (0–3)	0.019	2 (1–6)	1 (0–3)	0.022
Parity	1 (0–5)	0 (0–1)	0.006	1 (0–5)	0 (0–1)	0.006
IVF indication			Na			Na
Primary infertility		8 (53.3)			7 (53.8)	
Secondary infertility		3 (20)			3 (23.1)	
Tubal factor		3 (20)			2 (15.4)	
Male factor		1 (6.7)			1 (7.7)	
ET		2 (1–3)	Na		2 (1–3)	Na
Missing data		3			3	
OHSS		10 (66.7)	Na		10 (66.7)	Na
Mild		1 (10)			1 (10)	
Moderate		4 (40)			4 (40)	
Severe		5 (50)			5 (50)	
History	14 (73.7)	2 (13.3)	<0.001	7 (70)	2 (15.4)	0.008
ICP in previous pregnancy	8 (42.5)	0		4 (40)	0	
Family history of ICP	3 (15.7)	0		1 (10)	0	
Contraception-induced cholestasis	3 (15.7)	1 (6.7)		3 (30)	1 (7.7)	
Miscarriage	2 (10.5)	0		0	0	
Stillbirth	2 (10.5)	0		1 (10)	0	
Cholelithiasis or cholecystectomy	4 (21.1)	0		3 (30)	0	
Viral hepatitis	2 (10.5)	0		1 (10)	0	
Familial hyperlipidemia or cholestasis	1 (5.3)	1 (6.7)		0	1 (7.7)	
Jaundice	6 (31.6)	3 (21.4)	0.518	3 (30)	3 (23.1)	0.793
ICP onset, GW	15.2 ± 8.0	8.1 ± 6.7	0.003	8.7 ± 2.7	5.8 ± 3.0	0.029
ICP recurrence ^a^	11 (61.1)	5 (45.5)	0.089	5 (55.5)	3 (33.3)	0.230
Peak LFT						
ALT, U/L	119 (35–2415)	275 (85–3372)	0.270	159 (95–1496)	314 (85–3372)	0.650
Missing data	4	1		3	1	
AST, U/L	123 (48–2748)	255 (1156)	0.240	123 (86–1117)	225 (109–1156)	0.672
Missing data	4	2		3	1	
T-Br, μmol/L	84.6 (31–170)	22.3 (6.8–161)	0.010	98.3 (47.8–114)	23.1 (6.8–161)	0.073
Missing data	9	6		6	5	
TBA, μmol/L	141.1 (12–462)	137.2 (18.3–308)	0.869	242 (27–462)	137.2 (18.3–308)	0.113
Missing data	2	4		1	4	
ICP treatment			0.439			0.412
UDCA	14 (73.7)	8 (53.3)		9 (90)	6 (46.2)	
Antihistamines	8 (42.1)	5 (33.3)		4 (40)	4 (30.8)	
Corticosteroids	5 (26.3)	1 (6.7)		1 (10)	1 (7.7)	
Cholestyramine	2 (10.5)	1 (6.7)		1 (10)	1 (7.7)	
Hypnotics	2 (10.5)	0		1 (10)	0	
Plasmapheresis	2 (10.5)	0		1 (10)	0	
SAMe	1 (5.3)	3 (20)		1 (10)	1 (7.7)	
Pregnancy outcomes						
Delivery	18 (94.7)	11 (73.3)	0.080	9 (90)	9 (75)	0.231
Miscarriage	0	4 (26.7)	0.017	0	4 (25)	0.054
Artificial abortion	1 (5.3)	0	0.367	1 (10)	0	0.599
GA at delivery, GW ^a^	35 (29–38)	36 (30–39)	0.769	35 (30–37)	37 (34–39)	0.868
Term delivery	4 (22.2)	4 (40)	0.439	1 (11.1)	4 (50)	0.412
Preterm delivery	14 (77.8)	6 (60)	0.035	8 (88.9)	4 (50)	0.053
Singleton pregnancy	17 (94.4)	7 (63.6)		9 (100)	6 (66.7)	0.365
Twin pregnancy	1 (5.6)	4 (36.4)		0	3 (33.3)	0.103
Livebirth	18	16		8	13	0.232
Stillbirth	1	0		1	0	0.244
Delivery mode ^a^			0.219			0.123
Spontaneous vaginal delivery	4 (22.2)	4 (36.4)		0	3 (37.5)	
Induced vaginal delivery	8 (44.4)	2 (18.2)		7 (77.8)	2 (25.0)	
Planned Caesarean section	3 (16.7)	1 (9.1)		1 (11.1)	1 (12.5)	
Urgent Caesarean section	3 (16.7)	3 (27.3)		1 (11.1)	2 (25.0)	
Obstetrical complications	*n* = 19	*n* = 12		*n* = 9	*n* = 8	
PPROM	9 (47.4)	3 (25)	0.097	5 (55.5)	1 (12.5)	0.022
LFT aberration	2 (10.5)	3 (25)	0.439	1 (11.1)	3 (37.5)	0.412
Fetal distress	1 (5.3)	1 (8.3)	0.863	0	1 (12.5)	0.370
MAF	2 (10.5)	1 (8.3)	0.694	1 (11.1)	0	0.244
Presentation	2 (10.5)	1 (8.3)	0.195	1 (11.1)	0	0.244
Maternal comorbidities	2 (10.5)	0	0.195	0	0	Na
Chorioamnionitis	1 (5.3)	0	0.367	1 (11.1)	0	0.244
Oligohydramnios	0	1 (8.3)	0.253	0	1 (12.5)	0.370
Doppler aberrations	0	1 (8.3)	0.253	0	1 (12.5)	0.370
Preeclampsia	0	1 (8.3)	0.253	0	1 (12.5)	0.370
Newborns with perinatal complication	6 (33.3)	1 (6.25)		4 (50)	0	
Perinatal complications	*n* = 7	*n* = 3	0.148	*n* = 6	*n* = 0	0.021
Respiratory distress	4 (57.1)	1 (33.3)	0.240	3 (50)	0	0.034
Neonatal jaundice	1 (14.3)	1 (33.3)	0.863	1 (16.7)	0	0.244
Neonatal infection	1 (14.3)	1 (33.3)	Na	1 (16.7)	0	Na
Intestinal stenosis	1 (14.3)	0	Na	1 (16.7)	0	Na

Data are presented as mean values ± standard deviation or median values with minimal and maximal values for continuous variables or as numbers (percent) for categorical variables. ICP—intrahepatic cholestasis of pregnancy; IVF—in vitro fertilization; ET—embryo transfer; OHSS—ovarian hyperstimulation syndrome; LFT—liver function test; AST—aspartate aminotransferase; ALT—alanine transaminase; T-Br—total bilirubin; TBA—total serum bile acids; UDCA—ursodeoxycholic acid; SAMe—S-adenosyl methionine; GW—gestational weeks; GA—gestational age; PPROM—preterm premature rupture of membranes; MAF—meconium amniotic fluid; Na—not available. ^a^ Pregnancies with delivery, excluding miscarriage or abortion.

## Data Availability

Data is unavailable due to privacy and ethical restrictions, however, it can be retrieved by request to corresponding authors.

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
