# Peer review of "Severe Early-Onset Intrahepatic Cholestasis of Pregnancy Following Ovarian Hyperstimulation Syndrome with Pulmonary Presentation after In Vitro Fertilization: Case Report and Systematic Review of Case Reports"

_life, 2024, doi:10.3390/life14010129_

Round 1

Reviewer 1 Report

Comments and Suggestions for Authors

In the article ‘Severe early onset intrahepatic cholestasis of pregnancy following ovarian hyperstimulation syndrome with pulmonary presentation after in vitro fertilization: case report and systematic review of case reports.’  It is a valuable study.  However, there are some shortcomings and questions.

1)There are a lot of abbreviations in this article. Please use the full name when an abbreviation appears firstly in this article. It is not friendly to read this article. For example, In the line 16-17:  Evidence show that it can be developed in the first trimester, more commonly after IVF procedures with presence of ovarian hyperstimulation syndrome (OHSS). Please provide the full name for IVF here.

2) In the line 18-19: Systematic literature search of case reports/studies published in MEDLINE database were performed, from until July 2023, using selected terms. The grammar of this sentence, which is difficult to be understood, is incorrect.

3) The quality of Fig1 is very poor. Please re-make the photographs and provide the figure legends.

4) Even it beyond the scope of this article, could you add the potential reasons or mechanisms, which lead to ICP and early onset ICP in the introduction part.

5) In the Fig2, the 1st trimester ICP is analyzed. However, why the 2nd  trimester ICP is not included.

Comments on the Quality of English Language

In the article ‘Severe early onset intrahepatic cholestasis of pregnancy following ovarian hyperstimulation syndrome with pulmonary presentation after in vitro fertilization: case report and systematic review of case reports.’  It is a valuable study.  However, there are some shortcomings and questions.

1)There are a lot of abbreviations in this article. Please use the full name when an abbreviation appears firstly in this article. It is not friendly to read this article. For example, In the line 16-17:  Evidence show that it can be developed in the first trimester, more commonly after IVF procedures with presence of ovarian hyperstimulation syndrome (OHSS). Please provide the full name for IVF here.

2) In the line 18-19: Systematic literature search of case reports/studies published in MEDLINE database were performed, from until July 2023, using selected terms. The grammar of this sentence, which is difficult to be understood, is incorrect.

3) The quality of Fig1 is very poor. Please re-make the photographs and provide the figure legends.

4) Even it beyond the scope of this article, could you add the potential reasons or mechanisms, which lead to ICP and early onset ICP in the introduction part.

5) In the Fig2, the 1st trimester ICP is analyzed. However, why the 2nd  trimester ICP is not included.

Reviewer 2 Report

Comments and Suggestions for Authors

Dear author’s 

I was pleased to review your article and i have the following comment’s:

For instance i propose you to establish the aim of your paper. You report a rare case of ICP in a patient undergoing ivf procedures.

The medication of this patient could increase the risk of ICP?

What new info brings your case to the existing literature?

This disease is relatively rare in the first trimester. Why the review was performed on 2nd and 3rd trimester pregnancy to?

Minor English and grammar edits are required.

Round 2

Reviewer 2 Report

Comments and Suggestions for Authors

Thank you for your response.

please check the article for pumctuation edits.